# Learning to predict cutting angles from histological human brain sections

**Christian Schiffer**[*1,2]                                          C.SCHIFFER@FZ-JUELICH.DE
**Luisa Schuhmacher**[*3]                                    LUISA.SCHUHMACHER@HHU.DE
**Katrin Amunts**[1,4]                                                  K.AMUNTS@FZ-JUELICH.DE
**Timo Dickscheid**[1,2]                                          T.DICKSCHEID@FZ-JUELICH.DE

[1] *Institute of Neuroscience and Medicine (INM-1), Research Centre Jülich, Germany*

[2] *Helmholtz AI, Research Centre Jülich, Germany*

[3] *Institute of Computer Science, Heinrich-Heine-University Düsseldorf, Germany*

[4] *Cécile & Oscar Vogt Institute for Brain Research, University Hospital Düsseldorf, Germany*

**Editors:** Under Review for MIDL 2021

## Abstract

Studying brain architecture at the cellular level requires histological image analysis of sectioned postmortem samples. We trained a deep neural network to estimate relative angles between the cutting plane and the local 3D brain surface from 2D cortical image patches sampled from microscopic scans of human brain tissue sections. The model allows to automatically identify obliquely cut tissue parts, which often confuse downstream texture classification tasks and typically require specific treatment in image analysis workflows. It has immediate applications for the automated analysis of brain structures, like cytoarchitectonic mapping of the highly convoluted human brain.

**Keywords:** cytoarchitecture, histology, deep learning, human brain

## 1. Introduction

Analyzing the human brain based on digitized series of histological sections is considered the gold standard for mapping of cytoarchitectonic brain areas, which are important indicators for functional modules. Regions where the local angle between the cutting plane and the normal vector of the brain surface deviates significantly from 0° (*oblique cuts*, Fig. 1, A+B) cannot be reliably identified using established approaches (Schleicher et al., 1998; Schiffer et al., 2021), since they do not capture the cortical layer structure in the 2D plane. We show that a deep neural network can be trained to predict the local cutting angle from 2D image patches sampled in the isocortex. This allows us to give specific treatment to obliquely cut regions in order to improve robustness and precision of downstream image analysis tasks.

## 2. Methods

We formulate angle prediction as a regression problem and train a convolutional neural network (CNN) to predict cutting angles for each image pixel. Training data is derived from the BigBrain dataset (Amunts et al., 2013), a 3D reconstructed adult human brain

---

[*] Contributed equally

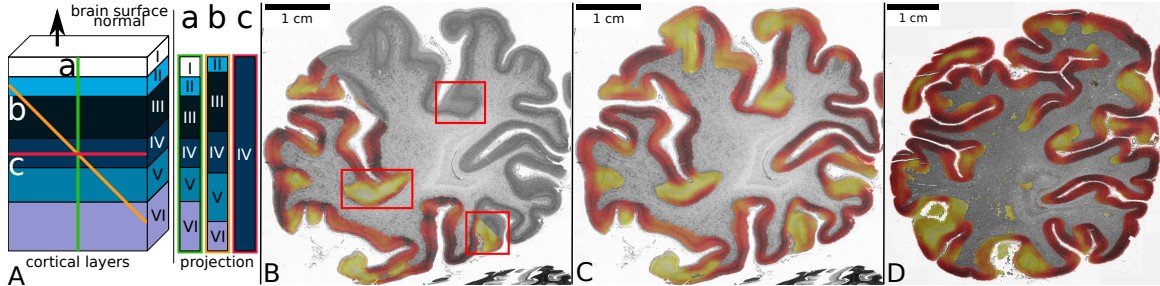

Figure 1: A) Illustration of the cortical layer structure. Cutting the cortex perpendicular to the brain surface (a) preserves the relative thickness of layers in the resulting projection, while more oblique cutting angles can lead to skewed layers (b) or complete loss of layer structure (c). B) Histological section 961 from the *BigBrain* dataset with color coded cutting angles (dark: close to 0°, bright: close to 90°). Boxes mark regions with high cutting angle, coinciding with locally increased cortical thickness. C) Cutting angles predicted by our model for the BigBrain section from B, and D) a section from a second brain which has undergone comparable histological processing steps. The model correctly predicts higher cutting angles in regions with locally increased cortical thickness in both brains.

created by aligning 7404 histological sections at 20 micron spatial resolution. The known 3D structure of this dataset is used to compute local angles between the 3D brain surface and the normal vector of the cutting plane for each voxel in the isocortex. Angles are then projected to the original histological sections from which the 3D BigBrain model was created. We use a model for pixel-wise regression from (Meyer et al., 2018). Model outputs are passed through a sigmoid function and scaled to the range of angles $\left[0, \frac{\pi}{2}\right]$. Training is performed on 384.000 image patches ($512^2$ pixel, 64µm per pixel) that we sample uniformly from the isocortex in 55 histological sections. Cortex pixels are identified using available segmentation masks for BigBrain (Lewis et al., 2014). The model is trained using Adam optimizer, learning rate 0.001, batch size 32 and mean squared error (MSE) loss. Cutting angles are only defined in the isocortex, so loss computation is restricted by available cortex segmentation masks. To reflect natural variations in the data, image patches are randomly mirrored and rotated by multiples of 90°. Model performance is quantitatively evaluated on 1000 image patches from 15 unseen histological sections from the BigBrain. We consider a pixel as correctly classified if the absolute difference $d_\alpha$ between predicted and real angle is below a threshold $t$ and compute pixel accuracy across increasing thresholds.

## 3. Results and discussion

We obtained accuracies 8.23% ($d_\alpha < 1$), 30.95% ($d_\alpha < 5$), 56.30% ($d_\alpha < 10$), and 84.64% ($d_\alpha < 20$). While accuracy is low for small thresholds, the model can reliably predict the cutting angle with a maximum error of 20°. This accuracy is sufficient to identify regions with a high cutting angle (e.g. greater than 60°), which is the key requirement for excluding obliquely cut regions from processing in analysis tasks. It should also be

noted that we cannot expect to reach 100% accuracy solely based on image patches, since the inversion of the local 2D projection is a highly underdetermined problem. Qualitative results (Fig. 1, C+D) from the BigBrain and a second brain which has undergone similar histological processing steps show that the model correctly identifies regions with locally increased cortical thickness as obliquely cut tissue. While no quantitative measures can be computed in the second brain due to lack of groundtruth data, visual inspection indicates that the model is well transferable to other subjects.

The presented work shows that it is possible to automatically identify obliquely cut tissue regions in histological sections using CNNs. Future work will investigate how this information can be used to improve performance of downstream tasks like cytoarchitecture mapping, e.g. by excluding oblique regions from the training set.

## Acknowledgments

This project received funding from the European Union's Horizon 2020 Research and Innovation Programme, grant agreement 945539 (HBP SGA3), and from the Helmholtz Association's Initiative and Networking Fund through the Helmholtz International BigBrain Analytics and Learning Laboratory (HIBALL) under the Helmholtz International Lab grant agreement InterLabs-0015. Computing time was granted through JARA on the supercomputer JURECA at Jülich Supercomputing Centre (JSC).

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
