# OpenReview forum: "Learning to predict cutting angles from histological human brain sections"
_MIDL.io/2021/Conference/Short — MIDL 2021 Poster_

### Official Review · Reviewer_HmyB · 2021-04-30

**Confidence:** 4
**Final Rating:** 2

**Summary:**


This paper demonstrates that deep learning can be useful to predict the cutting angle, i.e., the angle between the brain surface and cortical image patch. This task can be used to filter out bad-quality image patches for downstream tasks. The problem is posed as a regression task, and ground truth is computed from the known 3D structure of the brain. The authors used off-the-shelf CNN for this task which obtained performance ranging from 0.1 to 0.84 depending on the threshold of error tolerance in the prediction. Training/testing was done on image patches from different histological sections of a single brain, but authors have ensured no overlapping sections between the two. Only qualitative evaluation could be done on another brain.

**Strengths:**

- The paper proposes a new task and shows learnability with existing approaches. The proposed task can benefit other downstream tasks.

-  This task can be satisfactorily solved with existing techniques and expected to benefit downstream tasks.


**Weaknesses:**


- My main concern with this paper is that the authors did not report data or label distribution, nor did they have reported random/mean prediction baseline performance. Due to this, it is very hard to judge if the model has really learned anything or it is just predicting the mean label from the training set. It is likely that most cutting angles are near 0 (since one would mostly try to create "good" patches), and hence simply predicting something close to 0 may give high accuracy without learning anything. Further results and analysis are needed to understand if CNN has learned anything.

- The value of the task is due to the hypothesis that it will benefit the downstream task. The paper does not confirm this hypothesis nor present a very strong argument for this hypothesis. Mainly, how bad is an error of ~16% for downstream task performance is unclear.


**Deanonymize Review:**

no

**Detailed Comments:**

- See weaknesses

- Was the training set size 384 or 384,000? The use of the dot (384.000) in the paper makes it confusing.

- If the task is to identify the cutting angle is near 0 or not, why do authors pose this as a regression problem, and why not a 0-1 classification problem? I can see that by predicting the exact angles, the practitioner has to train a model once and decide what a suitable threshold for the downstream task is, but on the other hand, learning a classifier may be easier and more accurate. It would have been nice if the authors had provided some perspective about this in the paper.

- Why are results reported in terms of accuracy (by using thresholds) instead of metrics like MSE and MAE, which is usual for regression problems?

- The manuscript will benefit from a discussion of what other auxiliary information can improve performance since the authors mentioned that using only 2D patches cannot give 100% accuracy.



**Justification Of The Rating:**

The main argument of this paper that the downstream tasks can benefit from this type of filtering or quality control is unclear, which is ok given this is only a  short paper. However,  the quantitative information presented in the paper is not sufficient (missing random baseline, etc.) to judge if the neural network has learned anything non-trivial.

**Paper Type:**

validation/application paper

**Special Issue:**

no

---

### Official Review · Reviewer_AB5o · 2021-05-07

**Confidence:** 4
**Final Rating:** 3

**Summary:**

There is growing interest in investigating histological images with various microscopy techniques in order to study the brain. Authors propose to use a CNN based approach to estimate the cutting angle to study and analyze such data. They demonstrate promising quantitative results in their preliminary study.

**Strengths:**

- To the best of my knowledge, this is not a standing problem and use of CNN could be highly useful in addressing the need to estimate the cutting angle.
- Authors present sufficiently convincing quantitative evalution to demonstrate the validity of their approach.

**Weaknesses:**

- While there is quite detailed information about how the data used for training and validation, there is no information related to the CNN network.
- No comparison is provided against a baseline approach, which would be valuable in a longer version of this manuscript.

**Deanonymize Review:**

no

**Detailed Comments:**

- The manuscript is well-written. The organization of the text is logical and the use of language is adequate.
- In order to get valuable feedback on their approach, I recommend the authors to provide enough details about the used network architecture.
- In figure 1, the meaning of the colors from yellow to red is not clear.

**Justification Of The Rating:**

Authors proposed a CNN-based solution to a standing problem, which has good potential to work well. On the other hand, authors provided limited information about the methodological approach. Therefore, it is hard to comment on the validity of the technical part of the work. Overall, the presented results are indeed promising. Therefore, I think the MIDL community would find this work interesting and authors could also get useful feedback to improve their approach.

**Paper Type:**

methodological development

**Special Issue:**

no

---

### Meta-Review · Area_Chair_TZb7 · 2021-05-10

**Recommendation:** Accept (Poster)
**Confidence:** 4

**Metareview:**

The paper presents an interesting new application for standard deep learning methods (hence an application paper). The reviewers found the experiments of good value but made several suggestions for improving the presentation: reporting regression errors (or a distribution) adding some simple baselines. I believe these small improvements can be made in a final version and some space could be saved by abbreviating lengthy bibliography entries (you may use three authors plus et al. and use initials instead of full first names). I recommend acceptance.

---

### Decision · Program_Chairs · 2021-05-11

Accept (Poster)